# Intersecting Pathways: Treating Cocaine Withdrawal and Restless Leg Syndrome with Iron and Buproprion

**DOI:** 10.3390/healthcare12161570

**Published:** 2024-08-08

**Authors:** Sarah Hughes, Olivia Hill, Raja Mogallapu

**Affiliations:** School of Medicine, West Virginia University, Eastern Division, Martinsburg, WV 26506, USA; obhill@mix.wvu.edu

**Keywords:** cocaine withdrawal, iron-deficiency, dopamine

## Abstract

Many drastic actions are taken by cocaine users for the sake of experiencing high levels of dopamine, which depends on iron for its synthesis. Dopamine depletion and iron deficiency are also involved in the symptoms of restless leg syndrome (RLS). The intersecting biochemical pathways of cocaine use, iron deficiency, and RLS have not been adequately investigated. This case report reveals the successful treatment of a patient experiencing these conditions. A 63-year-old male with a history of cocaine use disorder, insomnia, and RLS sought emergency care for suicidality. Upon admission, he was also found to be iron deficient. He revealed that his RLS worsened when he attempted to abstain from cocaine. He also used alcohol to sustain the effects of cocaine when the cost of cocaine was too high. During hospitalization, his mood, cravings, and RLS were resolved with adjunctive iron supplementation, as well as treatment with 300 mg of Wellbutrin (bupropion hydroxychloride). If iron deficiency is present, the replenishment of the adequate dopaminergic receptor density and function via supplementation may play an essential role in the prevention of cocaine use and the cessation of cocaine withdrawal symptoms. Further research is warranted to validate these findings and to investigate the implications of iron supplementation in addiction medicine.

## 1. Introduction

Iron is one of the many cofactors involved in synthesizing monoamine neurotransmitters, including dopamine [1]. Iron deficiency can cause many health issues, including depression, headache, fatigue, and restless leg syndrome (RLS) [2,3]. Iron deficiency is also implicated in the downregulation of dopaminergic receptors, a process observed in chronic cocaine use [3,4]. As documented in the literature, stimulant use interferes with iron homeostasis, leading to iron deficiency in the caudate putamen, the nucleus accumbens, and the nigrostriatal tract, and iron accumulation in the globus pallidus and mesocortical tract [5,6]. Our patient is a 63-year-old male who presented with suicidal ideation, RLS, and chronic cocaine use. He was found to be iron deficient upon admission. The relationships between cocaine use, iron deficiency, and RLS appear to be significant due to their intersecting biochemical pathways and this patient’s worsening RLS when abstaining from cocaine. The significance of these relationships is exemplified by his symptom resolution after iron supplementation and ropinirole administration. This case report assesses the relationship between these factors and points to the possibility that addressing iron deficiency may decrease the risk of cocaine use and addiction.

## 2. Case Presentation

A 63-year-old male with a history of cocaine use disorder was admitted to the hospital for suicidal ideation and paranoia. He had not slept for the last five days, had been using alcohol and cocaine before his admission, and had been paranoid, with fleeting suicidal ideation, with no intent. The patient’s medical history included an abdominal aortic aneurysm (AAA), hypertension, and recurrent infections with the herpes simplex virus (HSV). His home medications were daily acyclovir, pantoprazole, and atorvastatin. Upon admission to the ED, a CBC revealed a reduced number of red blood cells and a high mean corpuscular volume (MCV). Additionally, a urine drug screen tested positive for cocaine. On his first day, the patient was very paranoid and minimally engaged in the conversation. The patient was closely monitored via his Clinical Institute Withdrawal Assessment (CIWA) score in the context of his alcohol use, with an initial score of 11, indicating minimal to mild withdrawal symptoms. A CIWA score aids clinicians in assessing the severity of alcohol withdrawal and in informing treatment, with scores ranging from 0–67. The patient’s level of suicidality was evaluated using an abbreviated Columbia Suicide Severity Rating Scale, on which he exemplified mild suicidality. The patient slept well after receiving 5 mg of Haldol (haloperidol decanoate) and 2 mg of Ativan (lorazepam) intramuscularly, medications which were only administered on this day.

On day two, the patient was tired but less paranoid. He denied any suicidal or homicidal ideations. On day three, the patient explained that he began using cocaine at the age of 30. He was sober from cocaine for ten years but had a recurrence of use two years prior to his hospitalization, with inpatient services being required for treatment of paranoia and decreased sleep. He explained that in the past 10 days, he had been increasing his cocaine use. He reported a significant strain on his family, financial difficulties, and several hospitalizations for cocaine use. He also confirmed that he experienced low motivation and a depressed and hopeless mood, triggering his cocaine cravings. Various treatment options were discussed, and he decided to start Wellbutrin (bupropion hydroxychloride) therapy to treat his lack of motivation and his cocaine use. He had used Wellbutrin in the past and felt that it was helpful for improving his motivation, but he had never used it in the context of treating his cocaine cravings. The patient was informed that the use of Wellbutrin is off-label for cocaine addiction treatment, and he expressed understanding of this fact after the benefits and side effects of Wellbutrin were discussed.

During his course in the hospital, the patient explained that whenever he stopped using cocaine, his RLS worsened, contributing to relapse. The patient tested positive for severe RLS symptoms on the RLS scale, with a score of 21–30. This scale is used by clinicians to assess the impact of RLS on a patient’s life and to determine the most effective treatment plan. While he was in the inpatient unit, the patient confirmed his occasional cravings for cocaine use, along with a depressed mood. Wellbutrin was titrated to 300 mg daily in the context of his depression, along with the initiation of 0.5 mg of Requip (ropinirole hydrochloride) daily for his RLS, which was eventually increased to 2 mg by the end of his hospital stay. The patient underwent iron labs and a complete blood count (CBC) as seen in Figure 1. His hemoglobin was 13.9 g/dL (14–18 g/dL), and his iron panel showed a ferritin level of 24 μg/dL (reference range 55–175 μg/dL) and an iron saturation of 9% (reference range of 15–50%). As the patient suffered from iron deficiency, a 325 mg ferrous sulfate tablet was prescribed daily. This particular preparation was chosen due to a lack of others on the hospital formulary at the time. An iron infusion was suggested during his stay in the hospital, but the patient refused it. Over the next few days, the patient reported improvement in sleep quality and mood, along with a reduction in cocaine cravings, and he became hopeful about his recovery. He also noticed a suppression of symptoms related to his RLS. He was discharged on day seven to a rehabilitation center for further stabilization of his cocaine use disorder. At this point in his admission, he had been on Wellbutrin for seven days, Requip for five days, and ferrous sulfate for four days total. In follow-up, he reported that with the continuation of his regimen, he had a sustained reduction in his cocaine cravings and was sober from substances. Upon initiation of this case report, written consent was obtained from the patient. 

## 3. Discussion

Dopamine synthesis requires several iron-binding enzymes, including tyrosine hydroxylase as illustrated in Figure 2 [2]. Dopamine is a neurotransmitter implicated in many physiological functions, including pleasure, memory, and addiction. Repeated cocaine use may lead to downregulated D2 dopaminergic receptors, as well as presynaptic dopamine release from the neurons. This disruption occurs primarily in the orbitofrontal cortex, cingulate gyrus, and dorsolateral prefrontal cortex, triggering behaviors such as impulsivity and impaired executive function. One technique to prolong the release of dopamine, as described above, is polysubstance use with alcohol and cocaine. When consumed together, the cocaine undergoes transesterification with ethanol to produce cocaethylene, which has a prolonged half-life and greater psychoactive effects than cocaine in isolation. Additionally, in this case, this phenomenon alleviated the impact of cocaine cessation, including the worsening of the patient’s RLS. Another reason that individuals use the two substances together is to alleviate the anxiety, depression, and pain that occur after the drug-induced high is over [7]. This polysubstance use warrants more attention from physicians, due to its popularity, high appeal, and numerous risks. 

In addition to altering dopamine physiology, cocaine use also impacts iron distribution. One study assessed 44 chronic cocaine users and the iron distribution throughout their bodies, finding that iron levels decreased significantly in the peripheries [5]. Another study demonstrated that iron deficiency has been associated with decreased dopamine receptors in rats, supporting the hypothesis that iron deficiency reduces the sensitivity of the dopamine receptors [6], stimulating cocaine withdrawal symptoms, and worsening RLS. Our patient likely experienced both conditions simultaneously, as his RLS was worse while attempting to stop cocaine use. This mechanism, along with the patient’s worsening RLS during cocaine cessation, indicates that iron deficiency and therefore, a reduced number and sensitivity of dopamine receptors, may increase the drive towards dopamine-producing substance use.

While cocaine users often have lower peripheral iron levels, they have been shown to exhibit higher iron levels in the globus pallidus [5,8]. One study showed that iron levels in the globus pallidus externus (GPe) were positively correlated with years of cocaine use [5]. Another study showed that iron levels in the globus pallidus internus (GPi) of cocaine users were much higher than those in the matched controls, and that these levels also increased with age; this is contrary to results for the normal aging process, these levels should instead decrease with age in the GPi [8]. More research is needed, but this research could provide more diagnostic information, as well as potential prognostic indicators and treatment targets, for cocaine users. 

Typically, oral iron replenishment takes up to several weeks, so we did not expect our patient’s RLS to completely resolve for some time [2]. However, his iron levels did go up two points in only two days, supporting iron supplementation as a source of his symptom improvement. Although ropinirole, a rapidly absorbed dopamine agonist, may have influenced the observed effects on his RLS, we believe that the addition of iron helped shorten the time to symptom improvement [9].

One systematic review suggested that fetal or infantile iron deficiency may lead to learning and memory deficits, as well as to increased emotional reactivity and mental health disorders. These findings may be associated with, and predispose individuals to, substance use disorders [10]. Identifying and treating our patient’s iron deficiency may be one of many factors that could have reduced his risk of cocaine addiction. Iron supplementation during cocaine withdrawal should thus be considered as an adjunctive therapy to help replenish dopaminergic stores.

When a patient elects to discontinue cocaine use abruptly, withdrawal symptoms will likely ensue. Anxiety, irritability, paranoia, and fatigue are common symptoms seen after cocaine cessation [11]. As described above, repeated cocaine use may decrease dopamine receptor sensitivity, leading to dopamine accumulation and prolonging addiction. Thus, many of the symptoms of cocaine withdrawal are associated with dopamine depletion. Any process that increases dopamine levels should help ameliorate cocaine withdrawal symptoms. Amantadine is an indirect dopamine agonist that may increase dopamine release [11]. A study by the National Institute of Drug Abuse assessed the use of amantadine for the treatment of cocaine withdrawal in a 61-participant trial. The study found that amantadine treatment improves abstinence in cases of very severe cocaine withdrawal. While our patient did not take Amantadine, the results of this study mechanistically support the theory that iron supplementation should also help with CUD.

Theoretically, any process that increases dopamine levels, such as treatment with Wellbutrin, Ropinerol, and iron supplementation, should ameliorate cocaine withdrawal symptoms. This was illustrated by our patient, who received Wellbutrin and iron supplementation, which are both interventions known to enhance dopamine production. Ropinirole, a rapidly absorbed dopamine agonist, was also used in the treatment plan and likely contributed to symptom relief by fully activating the D2 and D3 receptors [9]. The combined actions of ropinirole, amantadine, and iron on the dopamine levels in our patient likely worked synergistically to relieve his RLS. 

In this case, we also possess confounding variables, e.g., using Wellbutrin, which can potentiate decreased cravings. Wellbutrin prevents dopamine and norepinephrine reuptake, making it an appealing drug for substance use disorder treatment [12]. As a treatment for cocaine use disorder (CUD), Wellbutrin has shown mixed results in the literature. One retrospective cohort study showed that among 13 antidepressants used in treating CUD, Wellbutrin had the highest rates of CUD remission [13]. Another clinical trial showed that patients with ADHD and CUD exhibited significant improvements regarding their cravings and usage, while two other studies showed no differences between the placebo versus the treatment groups [14,15,16]. A recent study showed that Wellbutrin and simultaneous contingency management using iron supplementation resulted in improvement in CUD patients, while treatment with Wellbutrin alone did not [17]. While Wellbutrin alone may not have improved cocaine use disorder, its use with iron supplementation may have improved our patient’s cocaine cravings and withdrawal symptoms. 

Ropinirole, a commonly used medication for treating restless leg syndrome, may have influenced the observed effects, making it challenging to determine whether the outcomes were due to iron supplementation, the stimulation of dopaminergic receptors by ropinirole, or increased dopamine in the synapses achieved via Wellbutrin. Despite all of the above, our patient exhibited decreased cravings and remained sober for the last year after his RLS was addressed with Requip, Wellbutrin, and iron supplementation. This treatment method is an important avenue to explore, given the lack of treatment options for patients with stimulant use disorder. 

## 4. Conclusions

This case report demonstrates an example of iron deficiency, cocaine-use disorder, and RLS, three conditions with overlapping biochemical pathways that were successfully treated with adjunctive iron supplementation. If iron deficiency exists, the replenishment of adequate dopaminergic receptor density and function via supplementation may play an essential role in the prevention of cocaine use and the cessation of cocaine withdrawal. It is crucial to consider the potential of iron deficiency in patients who present symptoms of tiredness and lack of motivation, even in the absence of anemia. This consideration can lead to a more effective identification of the underlying causes and effective treatments, which might also help in the reduction of cocaine cravings. Furthermore, iron deficiency may perpetuate cocaine use and increase the severity of the addiction. This may demonstrate a need for an increase in screening for iron deficiency anemia as a primary preventative measure for substance use disorders.

It is important to acknowledge the confounding factors that may have improved the patient’s cocaine withdrawal symptoms, including the use of ropinirole, which is also a dopamine agonist. Further research is warranted to validate these findings and investigate the implications of iron supplementation in addiction medicine. It is crucial to explore the potential connection between iron deficiency, restless leg syndrome (RLS), and cocaine use disorder. Conducting additional case studies could provide valuable insights for better interventions and treatment of these conditions.

## Figures and Tables

**Figure 1 healthcare-12-01570-f001:**
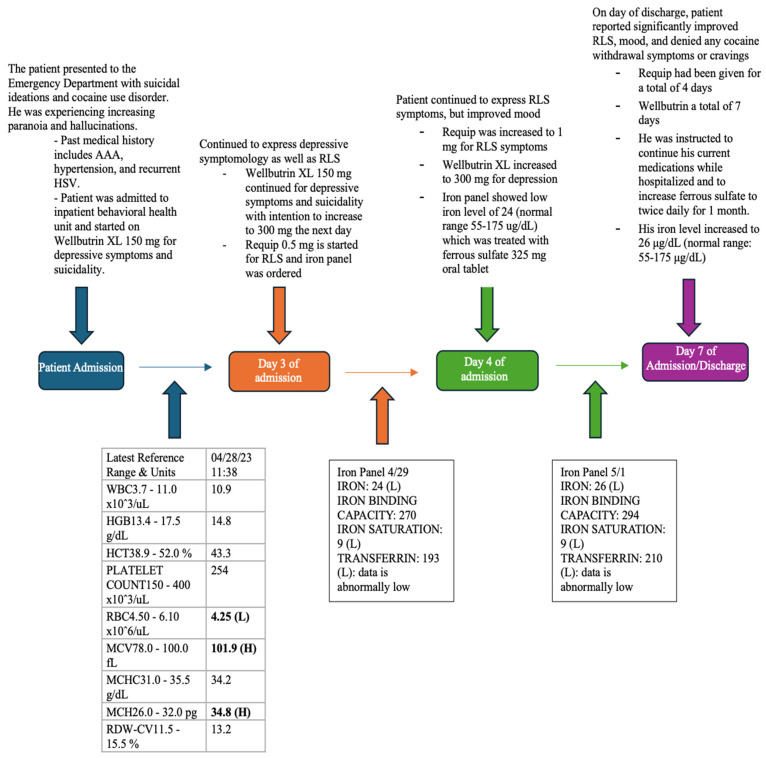
Timeline of Patient’s Inpatient Hospitalization: This timeline delineates key days throughout the admission period when medication adjustments or notable events took place, as well as presents the initial complete blood count (CBC) with abnormal results that support iron deficiency bolded in the table and iron panel results, along with a follow-up iron panel obtained two days later. The bolded data in the latest reference A final CBC was not performed prior to the patient’s discharge.

**Figure 2 healthcare-12-01570-f002:**
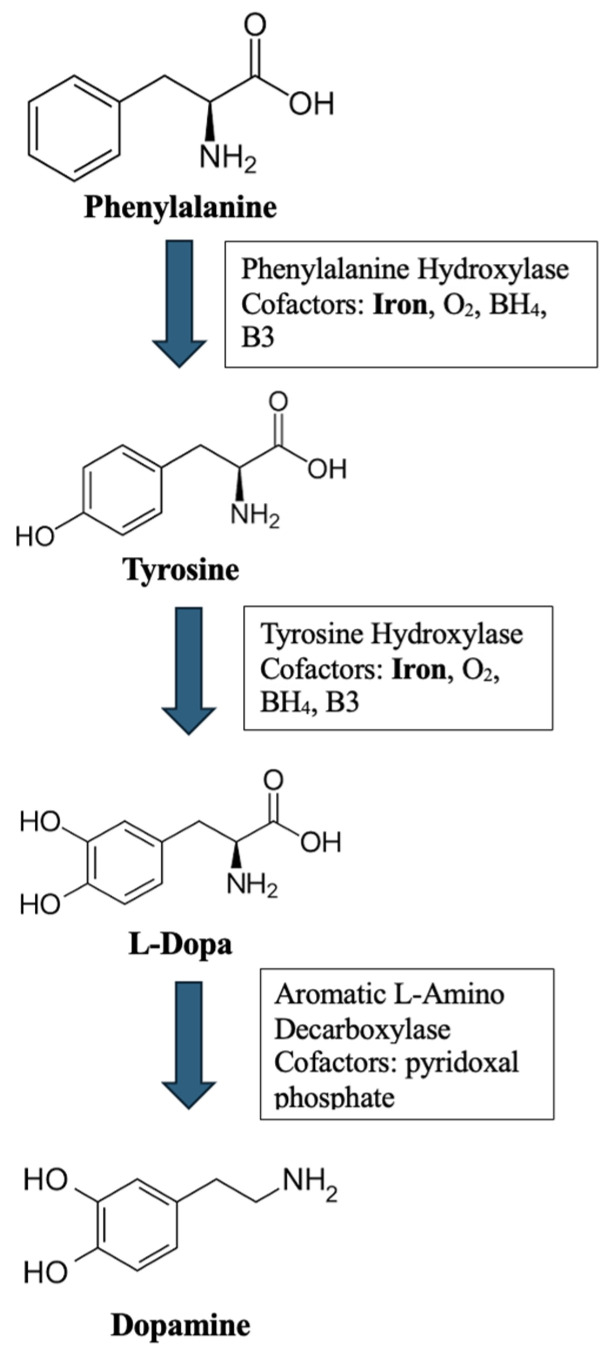
The Biosynthesis Pathway of Dopamine: This figure illustrates the biosynthesis pathway of dopamine from phenylalanine, illustrating the steps that require iron as a co-factor.

## Data Availability

The original contributions presented in the study are included in the article. Further inquiries can be directed to the corresponding author(s).

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
