# Peer review of "Intersecting Pathways: Treating Cocaine Withdrawal and Restless Leg Syndrome with Iron and Buproprion"

_healthcare, 2024, doi:10.3390/healthcare12161570_

Round 1
Reviewer 1 Report (Previous Reviewer 3)
Comments and Suggestions for Authors
Review of the paper entitled “Intersecting Pathways: Treating Cocaine Withdrawal and Restless Leg Syndrome with Iron and Buproprion” by Sarah Hughes, Olivia Hill and Raja Mogallapu
Currently, many researchers suggest a relationship between low iron levels and psychiatric disorders. This is explained by the fact that iron is an essential cofactor, along with oxygen and tetrahydrobiopterin for tyrosine hydroxylase activity, which is the rate-limiting enzyme for dopamine production.
The paper presented by the Authors describes the case of a 63-year-old man with a history of cocaine use disorder, insomnia, and restless legs syndrome (RLS). After admission to the hospital, the patient was found to be iron deficient. For this reason the patient received ferrous sulfate tablet in oral tablet form. In addition, the patient received Wellbutrin and Ropinirole in oral tablet form. The active substance of Wellbutrin is bupropion hydrochloride. Bupropion is a selective inhibitor of noradrenaline and dopamine reuptake. It also has a minimal effect on serotonin reuptake. Ropinirole, on the other hand, is an agonist of dopamine receptors.
The Authors conclude that during hospitalization, the patient's psychological well-being and RLS were resolved.
My comments
The topic discussed by the Authors is interesting. However, I believe that the paper presented by the Authors requires significant improvement.
· The Authors should first of all improve the Case Presentation section. The description should be arranged so that it creates a coherent, logical whole.
I suggest that the Authors divide the Case Presentation section into several subsections, for example:
Patient interview
What problem did the patient come in with? What is his past medical history? What medications is the patient currently taking?
The laboratory test and other tests performed on the patient after admission to the hospital
complete blood count and iron panel, Columbia Suicide Severity Rating Scale (C-SSRS), CIWA score and RLS scale
The treatment used
How long was the patient's hospitalization? For how many days was the patient treated with Haldol and Ativan; for how many days was the patient treated with Wellbutrin; for how many days was he treated with Ropinirole, and for how many days did he receive ferrous sulfate tablets? I suggest that the Authors present the patient's treatment course in the form of a timeline and a very short description.
The laboratory test and other tests performed on the patient on the last day of hospitalization
complete blood count and iron panel, Columbia Suicide Severity Rating Scale (C-SSRS), CIWA score and RLS scale
In my opinion the results of all laboratory and other tests after admission to the hospital and on the last day of hospitalization should be shown by the authors in the Table. The Authors conclude that during hospitalization, the patient's psychological well-being and RLS were resolved. The basis for the clinical assessment of the patient are laboratory tests and other tests, which should improve as a result of the treatment applied. Collecting these results in the Table will allow for their quick and easy comparison.
· I would like the Authors to comment on one more issue.
It is commonly known that RLS is associated with iron deficiency in patients. However, the literature data do not clearly indicate that iron therapy is an effective treatment for RLS. The systematic review (Cochrane Database of Systematic Reviews) that analyzed 428 people with RLS found that iron is better than a placebo for reducing the severity of RLS, although the benefit was low to moderate. The main measure of interest in this review was the severity of RLS. This was usually measured using a 10-question survey regarding severity and effects of urges to move the legs, called the IRLS (International Restless Legs Syndrome Severity Rating Scale). This was measured 2-4 weeks after injections of iron and 12-14 weeks after iron in pill form.
However, the Authors of the paper reviewed by me observed that RLS was resolved after just a few days after iron in pill form. This fact worries me.

Author Response
Please see the attachment.

Reviewer 2 Report (Previous Reviewer 2)
Comments and Suggestions for Authors
The manuscript was corrected.
Author Response
Comment 1: The manuscript was corrected.
Response 1: Thank you for your time and comment.
Round 2
Reviewer 1 Report (Previous Reviewer 3)
Comments and Suggestions for Authors
Review of the paper entitled “For Intersecting Pathways: Treating Cocaine Withdrawal and Restless Leg Syndrome with Iron and Buproprion” by Sarah Hughes, Olivia Hill and Raja Mogallapu
The Authors have revised their manuscript, making it much better. However, the basis for the clinical assessment of the patient are laboratory tests and other tests, which should improve as a result of the treatment applied. In response to my comment thus formulated, the Authors replied as follows: „Unfortunately, a final CBC was not taken for comparison before the patient’s discharge. Additionally, the scores such as CIWA and RLS that were recorded on admission were not followed up with additional follow-up or were not recorded by staff in the EMR”. This means that the Authors actually only have the iron panel results, along with a follow-up iron panel obtained two days later. These results have improved slightly. And this should be the final conclusion summarizing the patient's treatment. Meanwhile, the Authors state that the patient's sleep quality and mood improved, and a reduction in cocaine craving and symptoms associated with restless legs syndrome was observed. As I presume, the authors formulated these conclusions based on observations, conversations with the patient and the patient's subjective assessment. In the Authors' opinion, is such a significant improvement in the patient's condition the result of the patient's iron level going up two points in two days?
The “Timeline of Patient’s Inpatient Hospitalization” is very good. It can be supplemented with the results of iron binding capacity, iron saturation, transferrin and initial CBC, and then you can delete the Table. It is noteworthy that although the patient suffered from iron deficiency, CBC results were normal. Since the authors have few results, I propose combining the Case Presentation with the Discussion (with the consent of the Editor and the Editorial Office, of course). ). The discussion needs to be shorter, focusing on the results obtained, both laboratory and observational, i.e. those that the authors showed on the timeline.
It must be written in such a way as to convince the reviewer and editor that the authors' work, despite its limitations, is important. Strengths need to be clearly highlighted. You can also briefly describe the limitations.

Author Response
Please see the attachment.

This manuscript is a resubmission of an earlier submission. The following is a list of the peer review reports and author responses from that submission.
Round 1
Reviewer 1 Report
Comments and Suggestions for Authors
I
Extremely relevant and well described case. It should only clarify a few aspects, as described below:
- Identify the active substance of the medications used to treat the patient: Haldol, Ativan, Wellbutrin and Requip.
- Add title to figure 1.
- Was the case evaluated by an ethics committee? These aspects must be described in the text.
Author Response
For research article
|
Response to Reviewer X Comments
|
||
|
1. Summary |
|
|
|
Thank you very much for taking the time to review this manuscript. Please find the detailed responses below and the corresponding revisions/corrections highlighted/in track changes in the re-submitted files.
|
||
|
2. Questions for General Evaluation |
Reviewer’s Evaluation |
Response and Revisions |
|
Does the introduction provide sufficient background and include all relevant references? |
Yes/Can be improved/Must be improved/Not applicable |
[Please give your response if necessary. Or you can also give your corresponding response in the point-by-point response letter. The same as below] |
|
Are all the cited references relevant to the research? |
Yes/Can be improved/Must be improved/Not applicable |
|
|
Is the research design appropriate? |
Yes/Can be improved/Must be improved/Not applicable |
|
|
Are the methods adequately described? |
Yes/Can be improved/Must be improved/Not applicable |
|
|
Are the results clearly presented? |
Yes/Can be improved/Must be improved/Not applicable |
|
|
Are the conclusions supported by the results? |
Yes/Can be improved/Must be improved/Not applicable |
|
|
3. Point-by-point response to Comments and Suggestions for Authors |
||
|
Comments 1: Identify the active substance of the medications used to treat the patient: Haldol, Ativan, Wellbutrin and Requip. |
||
|
Response 1: Thank you for pointing this out. We agree with this comment. Therefore, we have identified the active substance in each of the medications used to treat our patient seen on page two, paragraph one, lines 51,52 for Haldol and Ativan. Page two, paragraph two, lines 65 and 66 for wellbutrin active ingredient. Page two, paragraph three, line 77 to address Requip. |
||
|
Comments 2: - Add title to figure 1. |
||
|
Response 2: Agree. We have, accordingly, added a title to Figure 1 on page 3, lines 96. Comments 3: Was the case evaluated by an ethics committee? These aspects must be described in the text. Response 3: Thank you for your comment. We have submitted this paper for review by an ethics committee. We, additionally, added a comment on page 2, paragraph 3, line 91 and 92 in regards to the written consent that was obtained prior to initiating this case report. The consent is attached below. |
||

Reviewer 2 Report
Comments and Suggestions for Authors
The present manuscript describes a case report of 63 year old male with a history of cocaine use disorder, insomnia and restless leg syndrome sought emergency care for suicidal. It is composed with requirements of the journal and written in proper style. However, in my opinion it can not be published in current form.
The title indicates that the Authors have taken up a very interesting and difficult topic: cocaine addiction. This is particularly important because no effective pharmacotherapy for this addiction is currently available, and all drugs used have only negligible effectiveness. However, the title definitely does not match the content of the manuscript. The Authors focused more on the restless leg syndrome and dependence on iron supplementation than on addiction therapy.
In the manuscript, after giving the trade names of the drugs, the Authors should put the name of the active substance in brackets, which will make it much easier to understand the described mechanisms. In the introduction, They should also include information about the importance of using the stimulant on iron level in structures other than the globus pallidus.
Ropinirole is a drug used quite often in the treatment of restless leg syndrome, so it is difficult to say whether iron supplementation actually influenced the observed effects or rather the stimulation of dopaminergic receptors by ropinirole.
The manuscript does not provide information on how long the patient was taking iron and why ferrus sulfate was chosen for this purpose. There are preparations available that are better absorbed, e.g. gluconate or chelated iron.
In the discussion, the Authors write that polydrug use is not well-studied, which is not true - there are numerous studies on the use of many substances at once and their effects.
In the discussion, the Authors write that iron definiciency decreases dopamine receptors and DAT activity. However, these actions have the opposite effect - while reducing the activity of receptors is responsible for the decrease in the effect of dopamine, reducing DAT activity increases its level. Please explain this phenomenon and its significance.
Line 126 does not have such a thing as dopamine sensitivity, perhaps it was about dopamine receptors sensitivity.
Line 132 The patient did not receive amantadine or this information is not included in the case report.
Line 157 The Authors claimed that manuscript demonstrate a need for an increase in screening for restless leg syndrome. However, they were mainly to deal with treatment of cocaine disorder by iron supplementation.
In the case report, too little attention was paid to the interaction of bupropion with cocaine. There are numerous studies on this subject and it is worth mentioning them, while showing possible interactions resulting from this combination and the effects of action. Especially since cocaine and bupropion share similar discriminative stimulus effects and some Authors suggest that bupropion may be a useful treatment agent in treatment-seeking cocaine abusers.
The Authors focused too much on the claim that iron supplementation led to the improvement of the patient's condition. The use of other substances, including dopaminergic receptor agonists (ropinirole), indicates something completely different. It is difficult to say whether supplementation had a positive effect.
Author Response
Thank you for your feedback. I have uploaded a word doc with our responses.

Reviewer 3 Report
Comments and Suggestions for Authors
Review of the paper entitled “Iron Supplementation in the Treatment of Cocaine Use Disorder”by Sarah Hughes, Olivia Hill and Raja Mogallapu
Interesting case report. However, I have a few suggestions.
The topic discussed by the Authors is very interesting. I have a few comments.
The approval of the Ethics Committee must be obtained.
The authors should explain the abbreviation CIWA. Moreover, I would ask the authors for a short description the CIWA scale. How many scores can measure the CIWA scale? What was the patient's result and how should it be interpreted?
I would also ask the authors for a similar, short description of the RLS scale?
The authors should explain the abbreviation CBC.
In my opinion the all laboratory and other tests (CIWA score, RLS scale) results during all days hospitalization should be shown by the authors in the Table (first day, second day, etc.).
On line 82 it says "Figures, Tables and Schemes". Meanwhile, in the entire manuscript there is only Figure 1 and no description. The authors present here a scheme of the dopamine synthesis pathway. The regulatory step of the dopamine synthesis pathway is the reaction catalyzed by tyrosine hydroxylase. It does so using molecular oxygen (O2), as well as divalent iron ions (Fe2+) and tetrahydrobiopterin (BH4) as cofactors. In the tyrosine hydroxylase reaction, BH4 is oxidized to dihydrobiopterin (BH2). BH4 is then regenerated by an NADPH dependent dihydrobiopterin reductase. Phenylalanine is hydroxylated at para-position by phenylalanine hydroxylase. It does so using molecular oxygen (O2), as well as divalent iron ions (Fe2+) and tetrahydrobiopterin (BH4) as cofactors. In the phenylalanine hydroxylase reaction, BH4 is oxidized to dihydrobiopterin (BH2). BH4 is then regenerated by an NADPH dependent dihydrobiopterin reductase. Thus, the hydroxylation reactions of phenylalanine and tyrosine occur according to the same mechanism. In my opinion, vitamin B9, i.e. folic acid, does not participate in the reaction catalyzed by tyrosine hydroxylase. Therefore, the scheme presented by authors should be corrected, taking into account my comment.
The authors should explain the abbreviation DAT.
Literature date have implicated elevated brain iron in conditions of prolonged psychostimulant exposure. It was, among others, demonstrated that individuals with cocaine use disorder (CUD) have excess iron in the globus pallidus internal segment (GPi). I would like the authors to comment on these data in the Discussion in the context of the results of their research presented in this paper.

Author Response
Thank you for your feedback. I uploaded a file with our response.
